# NISQ Algorithm for Hamiltonian simulation
# via truncated Taylor series

**Jonathan W. Z. Lau[1*], Tobias Haug[2], Leong C. Kwek[1,3,4], and Kishor Bharti[1†]**

**1** Centre for Quantum Technologies, National University of Singapore 117543, Singapore
**2** QOLS, Blackett Laboratory, Imperial College London SW7 2AZ, UK
**3** National Institute of Education, Nanyang Technological University,
1 Nanyang Walk, Singapore 637616
**4** MajuLab, CNRS-UNS-NUS-NTU International Joint Research Unit,
UMI 3654, Singapore

★ e0032323@u.nus.edu, † kishor.bharti1@gmail.com

## Abstract

Simulating the dynamics of many-body quantum systems is believed to be one of the first fields that quantum computers can show a quantum advantage over classical computers. Noisy intermediate-scale quantum (NISQ) algorithms aim at effectively using the currently available quantum hardware. For quantum simulation, various types of NISQ algorithms have been proposed with individual advantages as well as challenges. In this work, we propose a new algorithm, truncated Taylor quantum simulator (TQS), that shares the advantages of existing algorithms and alleviates some of the shortcomings. Our algorithm does not have any classical-quantum feedback loop and bypasses the barren plateau problem by construction. The classical part in our hybrid quantum-classical algorithm corresponds to a quadratically constrained quadratic program (QCQP) with a single quadratic equality constraint, which admits a semidefinite relaxation. The QCQP based classical optimization was recently introduced as the classical step in quantum assisted eigensolver (QAE), a NISQ algorithm for the Hamiltonian ground state problem. Thus, our work provides a conceptual unification between the NISQ algorithms for the Hamiltonian ground state problem and the Hamiltonian simulation. We recover differential equation-based NISQ algorithms for Hamiltonian simulation such as quantum assisted simulator (QAS) and variational quantum simulator (VQS) as particular cases of our algorithm. We test our algorithm on some toy examples on current cloud quantum computers. We also provide a systematic approach to improve the accuracy of our algorithm.



# 1  Introduction

Digital quantum computers have made immense progress in recent years, advancing to solving problems considered to take an unreasonable time to compute for classical computers [1,2]. In short, we are now in the Noisy Intermediate-Scale Quantum (NISQ) era [3,4], which is characterized by quantum computers with up to a few hundred noisy qubits and lacking full scale quantum error correction. Thus, noise will limit the usefulness of the computations carried out by these computers [3], preventing algorithms that offer quantum advantage for practical problems, such as Shor's algorithm for prime factorization [5], from being implemented.

However, just because such algorithms cannot be implemented on NISQ devices does not mean that quantum advantage for practical problems cannot be found with NISQ devices. There is currently great interest in the quantum computing and quantum information community to develop algorithms that can be run on NISQ devices but yet deal with problems that are practical [4,6,7]. Some of the most promising avenues deal with the problems in many-body physics and quantum chemistry. One major problem in this field is to develop algorithms capable of estimating the ground state and energy of many-body Hamiltonians. To such ends, algorithms like variational quantum eigensolver (VQE) [8,9] and quantum assisted eigensolver (QAE) [10,11] have been proposed.

The other major problem is to be able to simulate the dynamics of these many-body Hamiltonians. This task can be extremely challenging for classical computers, and Feynman proposed that this would be one of the areas where quantum computers could exhibit clear advantages over classical computers [12]. Powerful methods to simulate quantum dynamics on fault-tolerant quantum computers have been proposed, such as the concept of using truncated Taylor series by Berry et al [13].

On NISQ devices, a standard approach in simulating quantum dynamics is to utilize the Trotter-Suzuki decomposition of the unitary time evolution operator into small discrete steps. Each step is made up of efficiently implementable quantum gates, which can be run on the quantum computer [14–20]. However, the depth of the quantum circuit increases linearly with evolution time and the desired target accuracy. On NISQ devices, fidelity rapidly decreases after a few Trotter steps [21], implying long time scales will be unfeasible to simulate with this method. Alternatives to Trotterization have been proposed, such as VQS [22–24], subspace variational quantum simulator (SVQS) [25], variational fast forwarding (VFF) [26,27], fixed

state variational fast forwarding (fsVFF) [28], quantum assisted simulator [29,30] and generalized quantum assisted simulator (GQAS) [31] to name a few.

Recently, Otten, Cortes and Gray have proposed the idea of restarting the dynamics after every timestep by approximating the wavefunction with a variational ansatz [32]. Building on that, Barison, Vicentini and Carleo have proposed a new algorithm [33] for simulating quantum dynamics. Their algorithm, named projected variational quantum dynamics (pVQD) combines the Trotterization and VQS approaches [22,23]. They replace the differential equation with an optimization problem, although not well characterized, and require much simpler circuits compared to VQS. However, pVQD requires a quantum-classical feedback loop and might suffer from the barren plateau problem [34] as well the optimization problem may be computationally hard [35]. Further, the feedback loop mandates that one has to wait for each computation to finish before the next computation is run, which can be a major bottleneck on cloud-based quantum computers that are accessed via a queue.

Here, we propose the truncated Taylor quantum simulator (TQS) as new algorithm to simulate quantum dynamics. Our algorithm is building on the ideas of pVQD [32,33] combined with the ansatz generation of QAS [29], which we further enhance by applying the concept of truncated Taylor series by Berry et al [13]. Our contributions and our algorithm are as following:

1. We recast the simulation of the quantum dynamics as a quadratically constrained quadratic program (QCQP). This optimization problem, unlike the optimization problem in pVQD, is well characterized and invites rigorous analysis. The QCQP in our algorithm admits a semidefinite relaxation [10]. Moreover, based on ideas from [10], one can provide a sufficient condition for a local minimum to be a global minimum, which a solver can further use as a stopping criterion. Since the classical optimization program in QAE is also a QCQP, it helps us achieve a conceptual unification of TQS with QAE.

2. The differential equations which form the classical part of QAS and VQS can be recovered in our framework. Since VQS is already a particular case of QAS [29], our approach yields both VQS and QAS as special cases of TQS.

3. We remove the need for the classical-quantum feedback loop in pVQD. The absence of the feedback loop yields our algorithm to be exceptionally faster than the feedback loop based NISQ algorithms for simulating quantum dynamics such as [22, 25–28].

4. Our algorithm avoids the trainability issues that plague other variational quantum algorithms. The choice of a problem-aware ansatz and the structure of the TQS algorithm helps bypass the barren plateau problem. It is known that in variational quantum algorithms that rely on a parametric quantum circuit, there will always be a tradeoff between trainability and expressibility, implying that a highly expressible ansatz cannot be easily trainable [36]. In our case, we do not rely on parametric quantum circuits, thus we bypass this problem. Furthermore, our algorithm provides a systematic way to obtain a more expressible ansatz, which is missing in other algorithms.

## 2 TQS Approach

Let us first assume that the Hamiltonian $H$ is expressed as a linear combination of $r$ tensored Pauli matrices

$$H = \sum_{i=1}^{r} \beta_i P_i \,, \tag{1}$$

with coefficients $\beta_i \in \mathbb{C}$. The unitary evolution under the action of the Hamiltonian $H$ for time $\Delta t$ is given by

$$U(\Delta t) = \exp(-\iota H \Delta t) = \exp\left(-\iota \Delta t \sum_{j=1}^{r} \beta_j P_j\right) \tag{2}$$

$$= I - \iota \Delta t \left(\sum_{j=1}^{r} \beta_j P_j\right) - \frac{\Delta t^2}{2}\left(\sum_{j=1}^{r} \beta_j P_j\right)^2 + \mathcal{O}\left(\Delta t^3\right). \tag{3}$$

We do not need to implement the action of the unitary evolution in such a way. However, for purposes of describing the algorithm, we will use this power series expansion first, and talk more about alternatives later. We will now truncate the series, similar to [13]. If we choose small values of $\Delta t$ with respect to the eigen energies of $H$, we can approximate the unitary evolution with $V(\Delta t)$

$$U(\Delta t) \approx I - \iota \Delta t \left(\sum_{j=1}^{r} \beta_j P_j\right) \equiv V(\Delta t). \tag{4}$$

The classical evolution timestep $\Delta t$ should be chosen smaller than all relevant timescales of the Hamiltonian $H$ to be simulated. This requires knowledge of the spectrum of $H$, which in general is not available. However, we can find appropriate values for $\Delta t$ in an heuristic manner. In our algorithm, the evolution with $\Delta t$ is performed on a classical computer only and thus we can choose any value for $\Delta t$ without requiring any additional quantum computational cost. Thus, we can simply evolve with a very small value for $\Delta t$. To verify it is small enough, we can repeat the classical evolution for an even smaller value such as $\Delta t/2$. If the results for both $\Delta t$ and $\Delta t/2$ match, we can assume that $\Delta t$ provides sufficient accuracy.

Let us next choose the ansatz at time $t$ as linear combination of elements from cumulative $K$-moment states, $\mathbb{CS}_K$ (refer to [29] for the formal definition). These states are defined in the same way as in [29] and will be constructed with the help of the given Hamiltonian, by essentially considering powers of the Hamiltonian. In terms of Pauli matrices, given a set of $r$ tensored Pauli unitary matrices obtained from the unitary terms of the Hamiltonian $\mathcal{P} \equiv \{P_i\}_{i=1}^{r}$ and a positive integer $K$ and some efficiently preparable quantum state $|\psi\rangle$, the $K$-moment states are the set of quantum states of the form

$$\{|\chi\rangle\}_K = \{P_{i_K}\ldots P_{i_2}P_{i_1}|\psi\rangle\}_{i_K=1,\ldots,i_2=1,i_1=1}^{r}, \tag{5}$$

for $P_{i_l} \in \mathcal{P}$, where the indices $i$ all run from 1 to $r$. We note that we only include unique states within the set $\{|\chi\rangle\}_K$. This corresponds to removing any repeated Pauli unitary in $\mathcal{P}$. It should also be mentioned that the way the $K$-moment states are being generated is closely related to the Taylor expansion of the time evolution operator. If we consider the evolution of an arbitrary state by the time evolution operator, by observing that the Taylor expansion involves powers of the Hamiltonian $H$, it is clear that choosing the ansatz in such a way is suitable, as the $|\chi_i\rangle \in \{|\chi\rangle\}_K$ states are essentially states in the Hilbert space of $H^K|\psi\rangle$. This set is denoted by $\mathbb{S}_K$. The cumulative $K$-moment states $\mathbb{CS}_K$ are also defined in [29] as $\mathbb{CS}_K \equiv \cup_{j=0}^{K}\mathbb{S}_j$.

Now the ansatz is expressed as

$$|\psi(\alpha(t))\rangle_K = \sum_{|\chi_i\rangle \in \mathbb{CS}_K} \alpha_i(t)|\chi_i\rangle, \tag{6}$$

with some $\alpha_i \in \mathbb{C}$. For small values of $\Delta t$, the ansatz at time $t + \Delta t$ is given by

$$|\psi(\alpha(t+\Delta t))\rangle_K = \frac{V(\Delta t)|\psi(\alpha(t))\rangle_K}{\left(\langle\psi(\alpha(t))|_K V^\dagger(\Delta t) V(\Delta t)|\psi(\alpha(t))\rangle_K\right)^{\frac{1}{2}}}. \tag{7}$$

Using the ideas in [33], our goal now is to variationally approximate the time evolution of the system by adjusting our variational parameters. The crucial difference in our case is that our variational parameters $\alpha$ are coefficients which do not change the basis quantum states $|\chi_i\rangle$. Thus, they can be solely updated via a classical computer and do not require a quantum-classical feedback loop. To evolve by time $\Delta t$, we update the $\alpha_i$ parameters to $\alpha_i'$ such that the following fidelity measure is maximized

$$F\left(\alpha'\right) = \frac{\left|\langle\psi\left(\alpha'\right)|_K V\left(\Delta t\right)|\psi\left(\alpha\right)\rangle_K\right|^2}{\langle\psi\left(\alpha\right)|_K V^\dagger\left(\Delta t\right) V\left(\Delta t\right)|\psi\left(\alpha\right)\rangle_K}. \tag{8}$$

Using the notation $|\phi\rangle = V\left(\Delta t\right)|\psi\left(\alpha\right)\rangle_K$, the expression for fidelity becomes

$$F\left(\alpha'\right) = \frac{\langle\psi\left(\alpha'\right)|\phi\rangle_K \langle\phi|\psi\left(\alpha'\right)\rangle_K}{\langle\phi|\phi\rangle}. \tag{9}$$

Using the notation $W_\phi \equiv \frac{|\phi\rangle\langle\phi|}{\langle\phi|\phi\rangle}$, the above expression further simplifies to

$$F\left(\alpha'\right) = \langle\psi\left(\alpha'\right)|_K W_\phi|\psi\left(\alpha'\right)\rangle_K. \tag{10}$$

The goal is to maximize the fidelity subject to the constraint that $\langle\psi\left(\alpha'\right)|\psi\left(\alpha'\right)\rangle = 1$. Thus, the optimization program at timestep $t$ is given by

$$\max_{\alpha'} \ \langle\psi\left(\alpha'\right)|_K W_\phi|\psi\left(\alpha'\right)\rangle_K \tag{11}$$

$$\text{s.t.} \ \langle\psi\left(\alpha'\right)|\psi\left(\alpha'\right)\rangle_K = 1. \tag{12}$$

Using the elements from $\mathbb{CS}_K$ and the Hamiltonian $H$, we define the overlap matrices $\mathcal{E}$ and $\mathcal{D}$ as the following

$$\mathcal{E}_{m,n} = \langle\chi_m|\chi_n\rangle, \tag{13}$$

$$\mathcal{D}_{m,n} = \sum_j \beta_j \langle\chi_m|P_j|\chi_n\rangle. \tag{14}$$

Because of the way the $|\chi_n\rangle$ states are constructed, these values can be easily computed on a quantum computer, as they simplify to the expectation values of Pauli strings acting on the original quantum state $|\psi\rangle$. The constraint in the optimization program 12 can written in terms of $\alpha'$ as

$$\alpha'^\dagger \mathcal{E} \alpha' = 1. \tag{15}$$

We proceed to write the objective in the optimization program 12 in terms of the overlap matrices $\mathcal{E}$ and $\mathcal{D}$. In first order, we can simplify the expression

$$\begin{aligned}\langle\phi|\phi\rangle &= \langle\psi(\alpha)|_K \left(I + (\Delta t)^2 H^2\right)|\psi(\alpha)\rangle_K \\ &= \alpha^\dagger \mathcal{E} \alpha + O((\Delta t)^2) \approx \alpha^\dagger \mathcal{E} \alpha.\end{aligned} \tag{16}$$

Further, using the notation $G \equiv (\mathcal{E} - \iota\Delta t \mathcal{D})$ we find

$$\langle\psi\left(\alpha'\right)|\phi\rangle_K \langle\phi|\psi\left(\alpha'\right)\rangle_K = \alpha'^\dagger G \alpha \alpha^\dagger G^\dagger \alpha'. \tag{17}$$

Using Eq.15,16,17 and the notation $W_\alpha \equiv \frac{G\alpha\alpha^\dagger G^\dagger}{\alpha^\dagger \mathcal{E} \alpha}$, the optimization program in 12 can be re-expressed in terms of overlap matrices as

$$\max_{\alpha'} \ \alpha'^\dagger W_\alpha \alpha' \tag{18}$$

$$\text{s.t} \ \alpha'^\dagger \mathcal{E} \alpha' = 1. \tag{19}$$

The aforementioned optimization program is a quadratically constrained quadratic program with a single equality constraint. As described in [10], this QCQP admits a direct convex SDP relaxation. Moreover, the results from [10] provide a sufficient condition for a local minimum to be a global minimum, which a solver can further use as a stopping criterion. Alternatively, the problem can be solved with the classic Rayleigh-Ritz procedure by finding the eigenvector associated with the largest eigenvalue $\lambda$ of the generalized eigenvalue problem $W_\alpha \alpha' = \lambda \mathcal{E} \alpha'$.

It can be shown that in the limit of small $\Delta t$, TQS reduces to QAS (see Appendix C). This could potentially give us a way to obtain systematic higher-order corrections to the QAS matrix differential equation. Interestingly, this is a conceptual unification of the ground state problem (QAE) with the dynamics problem (QAS) in the quantum assisted framework. In QAE, finding the ground state and ground state energy of a Hamiltonian was formulated to become a QCQP. In TQS, the problem of simulating the dynamics is also given as a QCQP. This is conceptually satisfying as the problem of finding the dynamics is expressed as $e^{-itH} |\psi\rangle$, which is mathematically similar to using imaginary time evolution to finding the ground state via $e^{-\tau H} |\psi\rangle$. The aforementioned connection is also one of the primary justifications for ansatz selection in [11]. We note that as alternative it is possible implement the unitary evolution operator $U(\Delta t)$ directly instead of the Taylor series expansion of Eq.7, however this would require the usage of Hadamard tests (see Appendix D).

We want to emphasize again that the quantum computer is only required to measure the overlap matrices $\mathcal{E}$ and $\mathcal{D}$ at the start of the algorithm. No quantum-classical feedback loop for optimization is required. The only optimization steps required are performed solely on the classical computer with knowledge of the overlap matrices. The algorithm is as follows:

1. Choose an efficiently implementable initial state $|\psi\rangle$, then choose some K>0 and form the unique K-moment states $|\chi_i\rangle$ to construct the ansatz.

2. With knowledge of the Hamiltonian $H$, calculate the overlap matrices $\mathcal{E}$ and $\mathcal{D}$ on the quantum computer. The job of the quantum computer is now done.

3. Choose a small $\Delta t$ with respect to the eigenvalues of $H$ and evolve the state forward in time using a classical computer, by solving the optimization program 18 subject to the constraint 19.

If a higher fidelity for the simulation is desired, one can increase $K$ to acquire an ansatz with a higher expressibility. The timestep $\Delta t$ could be increased by including higher order terms in the power series expansion of $U(\Delta t)$ in our calculations (Described in Appendix E).

## 3 Results

We first use TQS to simulate a 2 qubit Heisenberg model

$$H_2 = \frac{1}{2}X_1 X_2 + \frac{1}{2}Y_1 Y_2 + \frac{1}{2}Z_1 Z_2. \tag{20}$$

We apply it to evolve an initial randomized 2 qubit state $|\psi_2\rangle$. This initial state is generated by 5 layers of $U_3$ rotations and CNOT gates on the 2 qubits (see Appendix A). We ran the TQS algorithm on the 5-qubit quantum computer *ibmq_rome*, available through IBM Quantum Experience. We used error mitigation by calibrating the measurement errors and applying a filter obtained from that calibration on our data with the toolbox provided in Qiskit [37]. The results are shown in Fig.1. The evolution of the state under TQS reproduces the exact behavior very well for an ansatz with $K = 1$ moment states, even in the presence of the noise of a real quantum computer.

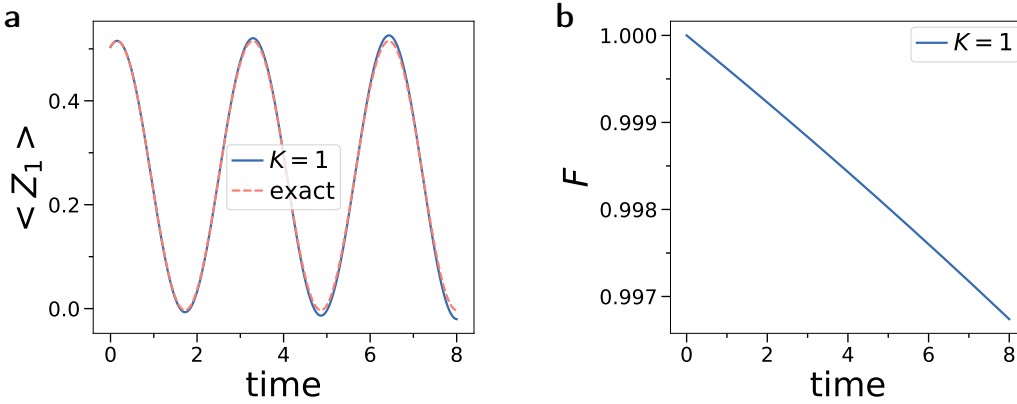

Figure 1: Time evolution of TQS on a 2 qubit state, with Hamiltonian $H_2$, simulated on the IBM quantum processor *ibmq_rome*. **a)** Expectation value of $\langle Z_1 \rangle$ **b)** Fidelity of the state.

Next, we apply TQS to simulate a 4 qubit XX chain model on a quantum computer

$$H_4 = \frac{1}{2}X_1 X_2 + \frac{1}{2}X_2 X_3 + \frac{1}{2}X_3 X_4. \tag{21}$$

Although this Hamiltonian is analytically solvable, we simulate this as a proof of principle. In Fig.2, we simulate (21) on *ibmq_rome* with an initial randomized 4 qubit state, generated by 5 layers of $U_3$ rotations and CNOT gates (see Appendix A). We run it for the $K = 1$ to $K = 3$ moment states. The evolution of the state under TQS again reproduces the exact behavior very well for the $K = 3$ case. We would like to point out that our algorithm can accurately simulate dynamics even for long time periods. The only errors that enter our algorithm are due to the ansatz being not expressible enough, and noise in the measurement of the matrix elements. Both type of errors affect only the initial conditions of the classical post-processing part. However, errors do not enter during the computation of the evolution itself as they are fully calculated on the classical computer. If we are able to obtain very accurate initial measurements for our matrix elements, and use an ansatz that fully captures the solution space, we believe that our algorithm in general will be able to simulate the dynamics accurately for long timescales.

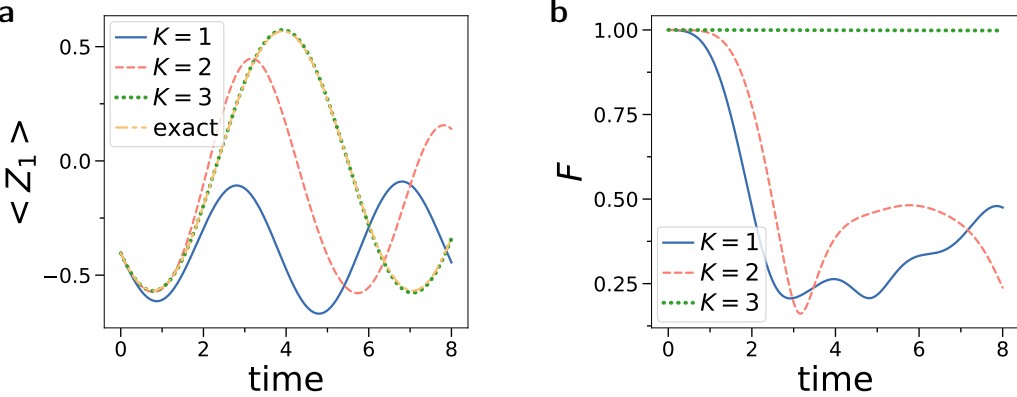

Figure 2: Time evolution of TQS on a 4 qubit state with Hamiltonian $H_4$ simulated on the IBM quantum processor *ibmq_rome*. **a)** Expectation value of $\langle Z_1 \rangle$ **b)** Fidelity with exact solution.

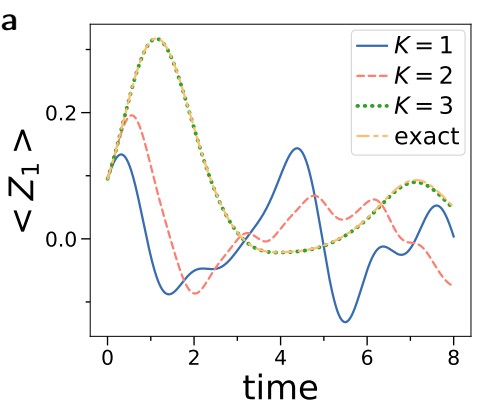
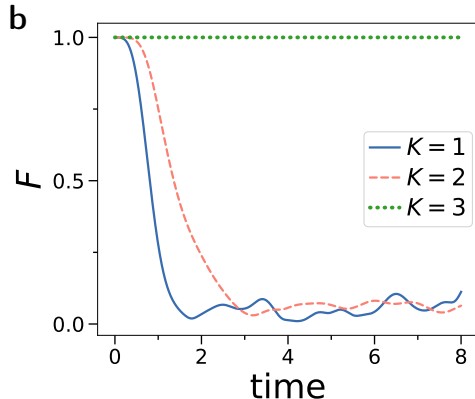

Figure 3: Time evolution of TQS on a 8 qubit state, with Hamiltonian $H_8$, simulated on a classical computer, with a random initial state. The initial state was generated with 3 successive layers of $U_3$ rotations with randomized parameters on each qubit, followed by CNOT/entangling gates. This is further described in Appendix A. **a)** Expectation value of $\langle Z_1 \rangle$. **b)** Fidelity of the state.

Next, we investigate in Fig.3 the transverse Ising model with 8 qubits by simulating TQS on a classical computer.

$$H_8 = \sum_{i=1}^{7} \frac{1}{2} Z_i Z_{i+1} + \sum_{j=1}^{8} X_j. \tag{22}$$

With an initial random state, we find that the evolution of the state reproduces the exact dynamics for the case of $K = 3$ moment expansion.

Lastly, we compare TQS to pVQD for a 2 qubit transverse Ising model on a simulation. We consider the 2 qubit transverse Ising Hamiltonian

$$H_{TFI,2} = \frac{1}{2} Z_1 Z_2 + \sum_{j=1}^{2} X_j. \tag{23}$$

We compare the algorithms with noisy simulators, where the noise models taken from the IBM Quantum Experience provider. The results are shown in Fig.4. While both TQS and pVQD show errors when simulating this Hamiltonian in the presence of noise, the expectation values for TQS are closer to the exact results most of the time. This is especially the case for the expectation value of $\langle Z_1 \rangle$. However, while the results might be argued to be somewhat similar, the resource requirements of both algorithms on the quantum computer are quite different. The TQS algorithm requires $\approx 30$ circuits to be run, while the pVQD simulator requires well over 4000 circuits, which is a major effort to run on the IBM Quantum Experience. We note that to increase the simulation time for this example, no extra circuits are required with TQS as the algebra already has closed, whereas the number of circuits in pVQD scales linearly with simulation time. Furthermore, TQS performs well with circuit that are shallower compared to pVQD, which requires a circuit with 8 variational parameters. This behavior of TQS requiring less variational parameters to get a similar result seems to be consistent for the small models we tested, as other variational algorithms usually need an over-parameterized ansatz to perform well.

We also compare TQS to Trotterization on a noisy simulator for the same 2 qubit transverse Ising Hamiltonian. A simple Trotterization of the time evolution operator for this case is

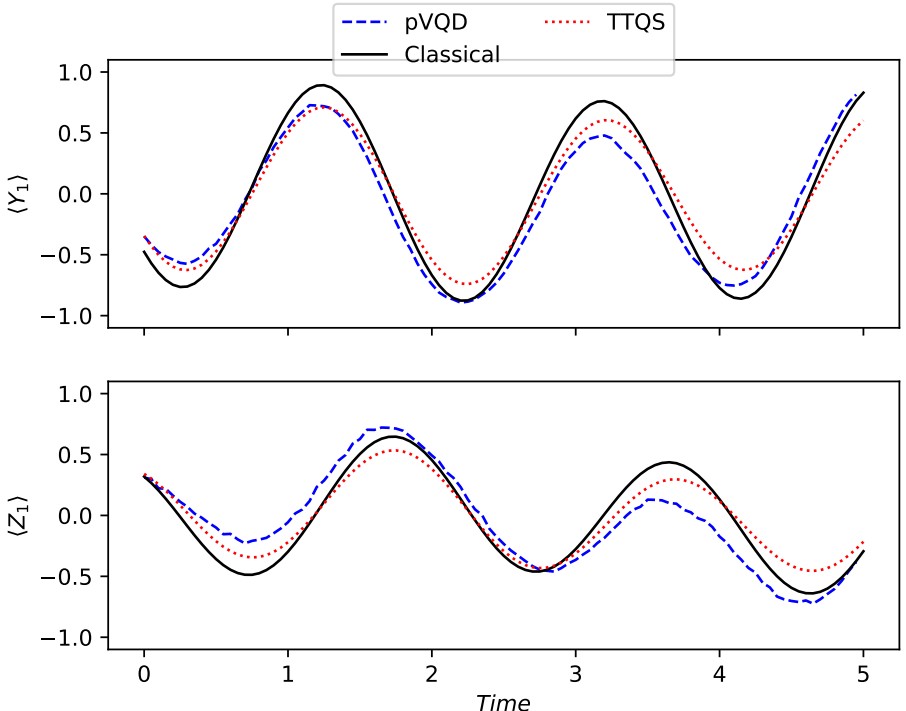

Figure 4: Time evolution of TQS and pVQD on a 2 qubit state, with Hamiltonian $H_{TFI,2}$, simulated with a noisy simulator. The noise model was taken from the IBM Quantum Experience provider, mimicking the noise of the real quantum processor *ibmq_bogota*. pVQD was run for 100 optimization steps, and made use of a parametric quantum circuit with 8 parameters, made out of successive layers of single qubit $X$ rotations and 2-qubit $ZZ$ rotations. The expectation values of $\langle Y_1 \rangle$ and $\langle Z_1 \rangle$ are plotted.

decomposed as

$$e^{-i\tau H_{2,TFI}} \approx \prod_{i=1}^{N} \left( \left( e^{-i\delta t_i Z_1 Z_2} \right) \left( e^{-i\delta t_i X_1} e^{-i\delta t_i X_2} \right) \right), \tag{24}$$

with $\sum_{i=1}^{N} \delta t_i = \tau$. The results are shown in Fig.5. As can be seen, even for a simple case such as this, due to the circuit lengths in Trotter increasing linearly with the time, the circuit lengths rapidly grow too long to obtain any meaningful results from the quantum computer. This is in contrast to TQS, which is able to capture the dynamics faithfully.

In Fig.6, we study our algorithm for up to thousands of qubits $N$. We use a Hamiltonian $H = \sum_{i=1}^{r} P_i$ that consists of $r$ randomly chosen $N$-body Pauli strings $P_i = \otimes_{j=1}^{N} \boldsymbol{\sigma}_j$, where $\boldsymbol{\sigma}_j \in \{I, X, Y, Z\}$. The cumulative $K$-moment states close at order $K = r$ and yield the full ansatz space necessary to describe the dynamics exactly. We use the product state $|0\rangle^N$ as initial state for the dynamics. This choice makes the dynamics tractable for classical computation. However, choosing an highly entangled initial state $|\psi\rangle$ would require a quantum computer to evaluate the overlaps. For such intractable states, our method provides a possible quantum advantage.

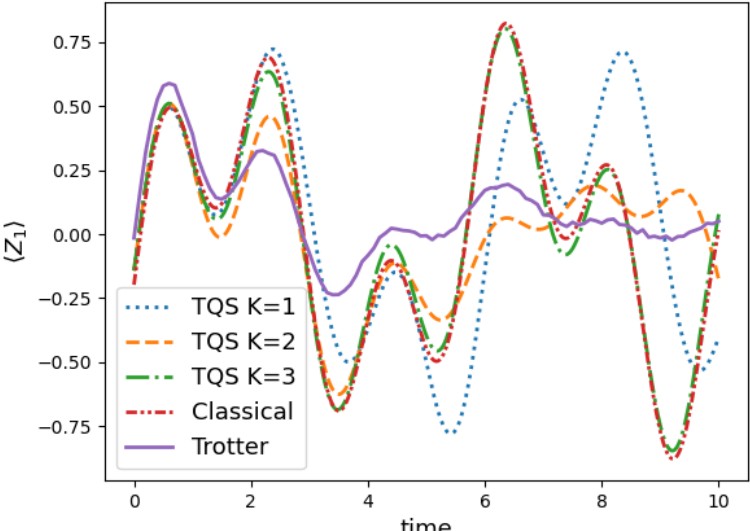

Figure 5: Dynamics of $H_{2,TFI}$ compared between Trotterization and TQS. The noise model was taken from the IBM Quantum Experience provider, mimicking the noise of the real quantum processor *ibmq_bogota*. We used a total of 100 steps for the Trotterized run. The expectation value of $\langle Z_1 \rangle$ is shown.

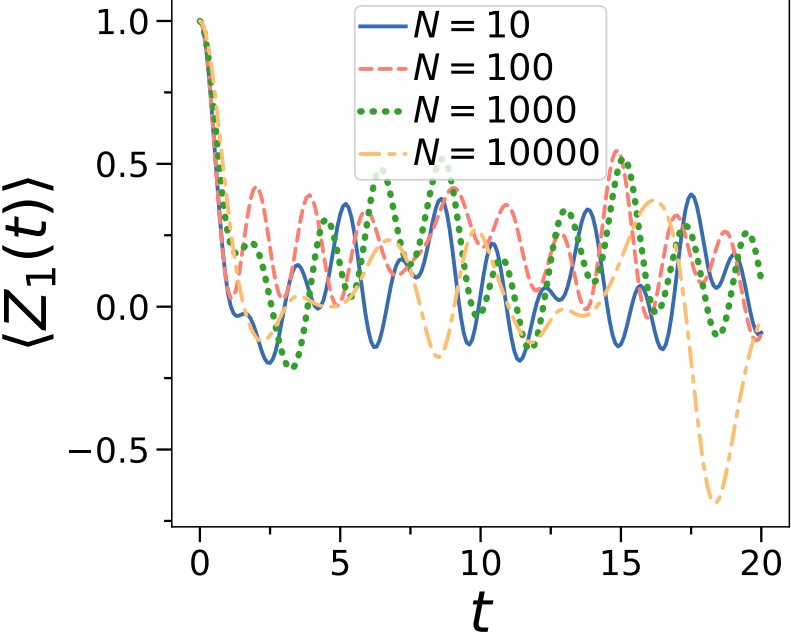

Figure 6: Dynamics of Hamiltonians consisting of multi-body Pauli strings for varying number of qubits $N$. Hamiltonians are composed of $r$ different random Pauli strings $H = \sum_{i=1}^{r} P_i$, where the Pauli strings $P_i = \otimes_{j=1}^{N} \sigma_j$ consist of $N$ tensored Pauli operators $\sigma_j \in \{I, X, Y, Z\}$. The initial state $|\psi\rangle = |0\rangle^{\otimes N}$ is the $N$-qubit product state with all zeros. The cumulative $K$-moment states consists of $2^r = 128$ ansatz states and exactly captures the full dynamics.

# 4 Discussion and Conclusion

The currently proposed NISQ algorithms face problems in scaling up to system sizes where classical computers cannot simulate the same systems, or in other words, to the point where we would see quantum advantage. For example, VQS/SVQS/pVQD require the use of a quantum-classical feedback loop, usually require complicated circuits, share similar problems as VQE like the barren plateau problem, and lack a systematic way to generate a parameterized ansatz. VFF and fsVFF also suffer from lacking a systematic way to generate the ansatz, usually require complicated circuits and have to run a quantum-classical feedback loop at the start. Further, the no fast-forwarding theorem suggests that not all Hamiltonians will be able to be accurately diagonalized with a reasonable amount of gates and circuit length, and the optimization step of the cost function in VFF might be too difficult to carry out efficiently. However, the barren plateau problem and ansatz state generation could be improved upon by applying various techniques [36, 38–41].

One problem that VQS and QAS share is that they require solving a differential equation which includes the pseudo-inverse of a matrix, whose elements are measured on a quantum computer. This matrix can be ill-conditioned. This procedure, via singular value decomposition, can be numerically unstable and sensitive to noise, especially as the system increases in size [42]. However, the sensitivity of these matrices has not been rigorously analyzed and more work has to be done to understand the scaling of the sensitivity.

In this work, we develop TQS for simulating quantum dynamics on digital quantum computers. TQS recasts the dynamical problem as a QCQP optimization program, which is well characterized unlike the optimization program in pVQD, allowing us to avoid the aforementioned problem in VQS and QAS.

At the same time, TQS retains the advantages of QAS, namely providing us a systematic method to select the ansatz, avoiding complicated Hadamard tests and controlled unitaries, avoiding the barren plateau problem, and only requiring usage of the quantum computer at the start, all of which are problems that are present in pVQD.

However, there are still many problems to tackle in our approach. One problem is an inherited problem from QAS. As the Hamiltonian size and complexity increase, large $K$ values may be needed to generate enough states for a sufficiently expressible ansatz to produce accurate results. It is clear from the connection between the Taylor expansion of the time evolution operator and our $K$-moment states that in the general case, the further in time we want to simulate, the exponentially larger our ansatz should be and the harder the difficulty of generating that ansatz. However, this is fundamentally a complexity theoretic statement which can not be bypassed in the general case by any quantum simulation algorithm based on parametric quantum circuits (variational quantum algorithms) or linear combination of quantum states (our algorithm). This problem particularly emerges in variational algorithms for time evolution. For example, in algorithms such as VQE for finding the ground state of Hamiltonians, we know that the ground state of locally gapped Hamiltonians fulfil area laws of entanglement and thus do not need exponentially many parameters to be described. However, for the time evolution over longer times a similar statement about the complexity of the problem is not known. Though our algorithm uses a problem aware ansatz, more information from the problem such as the combination coefficients $\beta_i$ and symmetries of the Hamiltonian could be employed to further tame the complexity. A further discussion and analysis on the number of states needed is given in Appendix B.

As the system size increases, it may be required to reduce $\Delta t$ to preserve accuracy in the classical post-processing part of the algorithm. This will increase the computational cost of the classical computer, however it requires no additional quantum computations. The number of classical optimization steps to be carried out increases linearly with the number of discretiza-

tions steps of the evolution time. Determining whether this poses a bottleneck for TQS when applied to large systems requires further studies.

Furthermore, in the presence of noise, the calculated fidelity of our states can go above one. A possible origin are small eigenvalues in the $\mathcal{E}$ overlap matrix, which can give the procedure of optimizing or solving the generalized eigenvalue problem numerical instability. As we scale up the system and consider more ansatz states, this issue can become more prevalent.

We expect our algorithm not to provide quantum advantage in the general case. However, we believe our algorithm is capable of providing quantum advantage over classical methods for certain cases. The conditions where we believe our algorithm will do so are:

- The basis states which are used to represent the initial quantum state are highly entangled such that they cannot be stored on a classical computer. This will render the calculation of corresponding overlaps classically hard, as it boils down to a circuit sampling task. Note that the Quantum Threshold Assumption (QUATH) by Aaronson and Chen [43] says that there is no polynomial-time classical algorithm which takes as input a random circuit $C$ and can decide with success probability at least $\frac{1}{2} + \Omega\left(\frac{1}{2^n}\right)$ whether $|\langle 0^n|C|0^n\rangle|^2$ is greater than the median of $|\langle 0^n|C|x^n\rangle|^2$ taken over all bit strings $x^n$. In other words, the circuit sampling task is difficult and hence classical algorithms will not be able to compete with algorithms based on circuit sampling as system size scales. The quantum part of TQS is based on circuit sampling which is classically difficult.

- The Hamiltonian possesses a particular structure. For example, the Hamiltonian consists of a small number of unitaries, the Krylov subspace closes fast, or the Hamiltonian is a low-rank matrix. We demonstrated such an example for a Hamiltonian consisting of a limited amount of multi-body Pauli strings where our method can simulate the dynamics of thousands of qubits. These Hamiltonians would be challenging for other methods such as Trotter or variational quantum algorithms. For those algorithms, the multi-body interactions and the large number of qubits would require an extensive number of gates and circuit depth to accurately represent the evolved state. A further example where our algorithm can perform well are quantum many-body scars. This quantum many-body phenomena can arise when the Krylov subspace closes fast at a low order $K$ [44], which is exactly the condition needed for our algorithm to perform well. The timescales that can be reasonably approximated by our algorithm is dependent on the Hamiltonian in question. Arbitrary Hamiltonians without the aforementioned conditions explore the full Hilbertspace during the evolution. Thus, it will be difficult for our ansatz to cover the whole solution space and approximate the dynamics accurately. Note that other variational quantum algorithms suffer similar problems as their ansatz is restricted to polynomial number of parameters. In the case of general Hamiltonians, our algorithm can provide systematic approximations for the quantum evolution of short time scales.

- The system size of interest and the amount of entanglement of the quantum state should be beyond the reach of classical simulation methods. Here, our algorithm can make use of the power of the quantum computer to prepare and measure classically intractable states.

In the future, the NISQ community should investigate these challenges, so that we can successfully run NISQ algorithms for larger qubit numbers.

## Acknowledgments

We are grateful to the National Research Foundation and the Ministry of Education, Singapore for financial support. The authors acknowledge the use of the IBM Quantum Experience devices for this work. This work is supported by a Samsung GRC project and the UK Hub in Quantum Computing and Simulation, part of the UK National Quantum Technologies Programme with funding from UKRI EPSRC grant EP/T001062/1.

## A   Details on running circuits on the IBM quantum computer

For the runs on the real quantum computer, we generated an initial state with randomized parameters to evolve with the following circuit. It comprised 5 layers of successive $U_3$ rotation with randomized parameters on each qubit, followed by a CNOT/entangling gate (see Fig.7 and 8). We sampled from each circuit with 8192 shots.

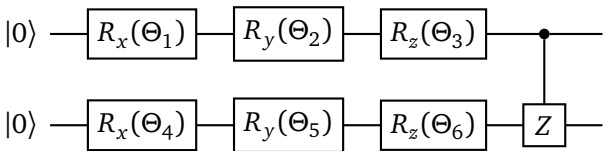

Figure 7: Circuit for two qubits that generate one set of $U_3$ rotation with randomized parameters, followed by a CNOT gate between the 2 qubits. 5 successive layers of this circuit were used to generate the initial starting state for the 2 qubit case on the IBM quantum computer for our runs of TQS. The $\Theta$s were randomly generated.

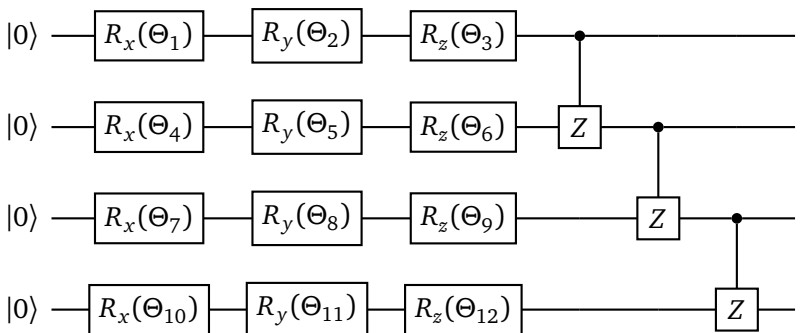

Figure 8: Circuit for four qubits that generate one set of $U_3$ rotation with randomized parameters, followed by a series of CNOT gates between the adjacent qubits. 5 successive layers of this circuit were used to generate the initial starting state for the 4 qubit case on the IBM quantum computer for our runs of TQS. The $\Theta$s were randomly generated.

## B   Number of basis states considered for each $K$, and discussion on scaling

The number of basis states that was used to construct the hybrid ansatz, for each $K$ moment expansion, for each Hamiltonian, is given in Table 1.

Table 1: Comparison of the number of basis states used to construct the hybrid ansatz for each $K$ for each Hamiltonian. For example, the $K = 2$ expansion for the 4 qubit case, using the Hamiltonian $H_4$, requires 4 quantum states to construct the hybrid ansatz. We only considered unique states, which correspond to only taking unique Pauli strings. For example, in the 8 qubit case, while the number of Pauli strings in the Hamiltonian is 15, which might suggest the $K = 3$ expansion generates $15^3 + 15^2 + 15 + 1 = 3616$ Pauli strings and thus 3616 states, many of them are repeated and only 137 of those strings are unique. Thus, we only end up having 137 states in our ansatz which turns out to be sufficient to represent the full dynamics of the $2^8 = 256$ dimensional Hilbertspace. This could be due to the transverse Ising model having underlying symmetries that reduce the number of basis states needed to capture the full dynamics.

|  | $K = 1$ | $K = 2$ | $K = 3$ | $K = 4$ |
|---|---|---|---|---|
| 2 Qubit Case | 1 | 4 |  |  |
| 4 Qubit Case | 1 | 4 | 7 | 8 |
| 8 Qubit Case | 1 | 17 | 137 |  |

Given a scalar $\tau$, an $N \times N$ matrix $A$ and an $N \times 1$ vector $v$, the action of the matrix exponential operator $\exp(\tau A)$ on $v$ can be approximated as

$$\exp(\tau A) v \approx p_{K-1}(\tau A) v, \tag{25}$$

where $p_{K-1}$ is a $K - 1$ degree polynomial. The approximation in equation 25 is an element of the Krylov subspace,

$$Kr_{K-1} \equiv span\left\{v, Av, \cdots, A^{K-1}v\right\}. \tag{26}$$

Thus, the problem of approximating $\exp(\tau A) v$ can be recast as finding an element from $Kr_K$. Note that the approximation in equation 25 becomes exact when $K - 1 = rank(A)$. In our case, we can identify $v$ with the initial state $|\psi\rangle$, $\tau$ with $-\iota t$ and $A$ with the Hamiltonian $H$.

In the worst case, the number of overlaps scales as $O(r^K)$ for $r$ terms in $H$. By observing the Taylor expansion of the time evolution operator $\exp(-iH\Delta t)$, we can see that at longer times we would struggle with finding an expressible enough ansatz in the general case, as we need to keep considering higher powers of $H$. This is fundamentally an expressibility problem, present in all NISQ variational algorithms, be it based on linear combination of states or those based on parametric quantum circuits. It is known that to prepare an arbitrary state on an $n$ qubit quantum computer, we require a circuit depth of at least $\frac{1}{n}2^n$ [45–48]. This suggests that it is very hard to produce an expressible enough Ansatz to reproduce an arbitrary quantum state in the Hilbert space.

It is known that the the Krylov subspace spans the entire space when you exponentiate the Hamiltonian $H$ to the power of $K - 1$, where $K - 1 = rank(H)$. Thus, the number of states that we require in our Ansatz scales linearly with the rank of the Hamiltonian.

Furthermore, one of the major contributions of the TQS algorithm is that, by using this problem-aware Ansatz, it provides a systematic way to obtain a more and more expressible Ansatz. The other variational algorithms like VQS and VFF still do not have a systematic method to generate an expressible enough Ansatz, or to improve on an Ansatz in a efficient way. Also, it has been shown that if we use a hardware efficient Ansatz, we would in general expect to encounter the barren plateau problem, which makes it very hard for the algorithm to train and optimize [34, 49]. Furthermore, the usual technique of using more and more layers of hardware efficient Ansatz circuits gives no guarantee that it will become more and more expressible in an efficient manner, when compared to the number of variational parameters that

we are adding. There is also no guarantee that this will indeed improve the appropriateness of the Ansatz. This is especially true for larger systems. In TQS, with the way we generate the Ansatz with $K$ moment states, we can see that at worst, we get an Ansatz with as many states as the size of the Hilbert space, which is fully expressible. This is due to the group of Pauli strings closing on itself eventually. Also, we can see that as we increase the $K$, we will definitely improve our Ansatz and get to a point where it is eventually expressible enough. In future, using the coefficients of the terms in the Hamiltonian, we expect to be able to slow down the growth of the number of states.

Our algorithm also relies on being able to calculate expectation values of powers of the Hamiltonian, $\langle \psi | H^k | \psi \rangle$ in an efficient manner. If we look at the Pauli string level (break our Hamiltonian into linear sums of Pauli strings), the number of Pauli terms in $H^k$ grows exponentially in k. Right now, for current implementation of our algorithm on available quantum computers, this breaking into Pauli strings is necessary due to the imperfections in said quantum computers. However, if we allow more complex operations that cannot be performed very well right now, such as complex controlled unitaries, the resources needed to measure such $\langle \psi | H^k | \psi \rangle$ values might scale less [50].

We would also like to mention that depending on the Lie algebra of the Pauli terms in the Hamiltonian and the rank of the Hamiltonian, the number of required overlaps can be a lot smaller compared to the upper bound. By considering specific kinds of Hamiltonians, the number of states needed will be manageable. As an example, for a system size with a multiple of 3 qubits, if we consider the Hamiltonian of the form $H = XYZXYZ...XYZ + YZXYZX...YZX + ZXYZXY...ZXY + XXXXX...XXX$, the set of $K$-moment states is maximally size 8, implying that 8 ansatz states are sufficient to simulate the dynamics with our algorithm.

## C  QAS and VQS as special cases of TQS

In this appendix, we show that in the limit of choosing a very small $\Delta t$, one obtains QAS from TQS. Since VQS is a special case of QAS [29], we get VQS also as special case of TQS. We start out with the series expansion of $|\psi(\vec{\alpha} + \delta\vec{\alpha})\rangle$

$$|\psi(\vec{\alpha} + \delta\vec{\alpha})\rangle = |\psi(\vec{\alpha})\rangle + \sum_j \frac{\partial}{\partial \alpha_j} |\psi(\vec{\alpha})\rangle \, \delta\alpha_j. \tag{27}$$

Now in TQS we want to maximize the overlap of $U(\Delta t)|\psi(\vec{\alpha})\rangle$ and $|\psi(\vec{\alpha} + \delta\vec{\alpha})\rangle$, which is essentially the fidelity measure in equation 8

$$
\begin{aligned}
&|\langle \psi(\vec{\alpha}) | U^\dagger(\Delta t) | \psi(\vec{\alpha} + \delta\vec{\alpha})\rangle|^2 \\
&= \left[ \langle \psi(\vec{\alpha})| U^\dagger(\Delta t)|\psi(\vec{\alpha})\rangle + \sum_j |\psi(\vec{\alpha})\rangle U^\dagger(\Delta t) \frac{\partial |\psi(\vec{\alpha})\rangle}{\partial \alpha_j} \delta\alpha_j \right] \times [\text{C. C.}] \\
&\overset{|\psi(\vec{\alpha})\rangle = \sum_j \alpha_j |\chi_j\rangle}{=} \left[ \langle \psi(\vec{\alpha})|U^\dagger(\Delta t)|\psi(\vec{\alpha})\rangle + \sum_j |\psi(\vec{\alpha})\rangle U^\dagger(\Delta t)|\chi_j\rangle \delta\alpha_j \right] \times [\text{C. C.}] \\
&= |\langle \psi(\vec{\alpha})|U^\dagger(\Delta t)|\psi(\vec{\alpha})\rangle|^2 + \sum_j \langle \psi(\vec{\alpha})|U^\dagger(\Delta t)|\chi_j\rangle \langle \psi(\vec{\alpha})|U(\Delta t)|\psi(\vec{\alpha})\rangle \delta\alpha_j \\
&\quad + \sum_j \langle \chi_j|U(\Delta t)|\psi(\vec{\alpha})\rangle \langle \psi(\vec{\alpha})|U^\dagger(\Delta t)|\psi(\vec{\alpha})\rangle \delta\alpha_j^* \\
&\quad + \sum_{j,k} \langle \psi(\vec{\alpha})|U^\dagger(\Delta t)|\chi_j\rangle \langle \chi_k|U(\Delta t)|\psi(\vec{\alpha})\rangle \delta\alpha_j \delta\alpha_k^*.
\end{aligned}
\tag{28}
$$

Now in the same manner as QAS, using the Mclachlan's variational principle [23, 29, 30, 51], we demand that the variation of this fidelity is equal to 0 with respect to $\alpha_j$:

$$\implies \langle\psi(\vec{\alpha})|U^\dagger(\Delta t)|\chi_j\rangle\langle\psi(\vec{\alpha})|U(\Delta t)|\psi(\vec{\alpha})\rangle + \sum_k \langle\psi(\vec{\alpha})|U^\dagger(\Delta t)|\chi_j\rangle\langle\chi_k|U(\Delta t)|\psi(\vec{\alpha})\rangle\delta\alpha_k^* = 0$$

$$\implies \langle\psi(\vec{\alpha})|U(\Delta t)|\psi(\vec{\alpha})\rangle + \sum_k \langle\chi_k|U(\Delta t)|\psi(\vec{\alpha})\rangle\delta\alpha_k^* = 0. \tag{29}$$

Now we substitute in $U(\delta t) = I - i\Delta t H$:

$$\implies \langle\psi(\vec{\alpha})|\psi(\vec{\alpha})\rangle - i\Delta t\langle\psi(\vec{\alpha})|H|\psi(\vec{\alpha})\rangle$$
$$+ \sum_k \langle\chi_k|\psi(\vec{\alpha})\rangle\delta\alpha_k^* - i\Delta t\sum_k \langle\chi_k|H|\psi(\vec{\alpha})\rangle\delta\alpha_k^* = 0. \tag{30}$$

Now we take the derivative of this equation with respect to $\Delta t$. Note that $\frac{d}{d\Delta t}\delta\alpha_k^* = \delta\dot{\alpha}_k^*$. We then discard any terms remaining that are linear in $\Delta t$ or in $\delta\alpha$ (implying we have chosen such a small $\Delta t$ that $\delta\alpha$ is also very small).

$$\implies -i\langle\psi(\vec{\alpha})|H|\psi(\vec{\alpha})\rangle + \sum_k \delta\dot{\alpha}_k^*\langle\chi_k|\psi(\vec{\alpha})\rangle\delta\alpha_k^* = 0. \tag{31}$$

Using the above definition of the $\mathcal{E}$ and $\mathcal{D}$ matrices in equation 13 and 14, this simplifies to:

$$\implies -i\vec{\alpha}^\dagger\mathcal{D}\vec{\alpha} + \vec{\alpha}^\dagger\mathcal{E}\vec{\alpha} = 0$$
$$\implies \mathcal{E}\vec{\alpha} = -i\mathcal{D}\vec{\alpha}. \tag{32}$$

This is exactly the same differential equation that we aim to solve in QAS. If we do not ignore the higher order terms, we could obtain systematic higher order corrections to the QAS matrix differential equation using such a method.

## D  Unitary implementation

As alternative, we could implement the unitary evolution operator $U(\Delta t)$ directly instead of the Taylor series expansion of Eq.7

$$|\psi(\alpha(t+\Delta t))\rangle_K = U(\Delta t)|\psi(\alpha(t))\rangle_K. \tag{33}$$

and defining the matrix $\mathcal{R}_{m,n} = \langle\chi_m|U(\Delta t)|\chi_n\rangle$ to solve the program

$$\max_{\alpha'} \alpha'^\dagger\mathcal{R}\alpha\alpha^\dagger\mathcal{R}^\dagger\alpha' \tag{34}$$

$$\text{s.t } \alpha'^\dagger\mathcal{E}\alpha' = 1, \tag{35}$$

$U(\Delta t)$ could be implemented with a Trotter decomposition or with an oracle. However, this complicates the circuits needed to calculate the $\mathcal{R}$ matrix, requiring the usage of Hadamard tests.

## E  Higher order approximations

We investigate higher order expansion for the evolution operator in this section. First, we define the overlap matrix $\mathcal{J}$

$$\mathcal{J}_{m,n} = \sum_{i,j}\beta_i\beta_j\langle\chi_m|P_iP_j|\chi_n\rangle. \tag{36}$$

Considering the next highest power expansion of $U(\Delta t)$:

$$U(\Delta t) \approx I - \iota\Delta t \left(\sum_{j=1}^{r} \beta_j P_j\right) - \frac{\Delta t^2}{2} \left(\sum_{j=1}^{r} \beta_j P_j\right)^2 \equiv V_2(\Delta t), \tag{37}$$

and defining $|\phi\rangle = V_2(\Delta t)|\psi(\alpha)\rangle_K$, the constraint in the optimization program 12 turns out to be still the same as equation 15:

$$\langle\psi(\alpha^{\dagger})|\psi(\alpha'^{\dagger})\rangle = \alpha'^{\dagger}\mathcal{E}\alpha'. \tag{38}$$

It turns out that $\langle\phi|\phi\rangle$ is actually exactly equal to $\alpha^{\dagger}E\alpha$, which is the result we used earlier in equation 16, as all the 2nd order terms nicely cancel out.

Now, using the notation $G_2 \equiv \left(\mathcal{E} - \iota\Delta t\mathcal{D} - \frac{\Delta t^2}{2}\mathcal{J}\right)$,

$$\langle\psi(\alpha')|\phi\rangle_K\langle\phi|\psi(\alpha')\rangle_K = \alpha'^{\dagger}G_2\alpha\alpha^{\dagger}G_2^{\dagger}\alpha'. \tag{39}$$

Now the optimization program in 12 can be re-expression in this higher order approximation as

$$\max_{\alpha'} \alpha'^{\dagger}\left(\frac{G_2\alpha\alpha^{\dagger}G_2^{\dagger}}{\alpha^{\dagger}\mathcal{E}\alpha}\right)\alpha' \tag{40}$$

$$\text{s.t } \alpha'^{\dagger}\mathcal{E}\alpha' = 1.$$

And using the notation $W_{2,\alpha} \equiv \frac{G_2\alpha\alpha^{\dagger}G_2^{\dagger}}{\alpha'\mathcal{E}\alpha}$, we further condense the above optimization program as

$$\max_{\alpha'} \alpha'^{\dagger}W_{2,\alpha}\alpha' \tag{41}$$

$$\text{s.t } \alpha'^{\dagger}\mathcal{E}\alpha' = 1. \tag{42}$$

Once again, the only work that the quantum computer need to do is to calculate overlap matrices in the start, in this case having to calculate $\mathcal{E}$, $\mathcal{D}$ and $\mathcal{J}$. In fact, when going from lower order approximations to higher order approximations, you can reuse the saved matrices and only calculate the new ones needed. In this case, in the original TQS, which uses a first order approximation for $U(\Delta t)$, we already have the $\mathcal{E}$ and $\mathcal{D}$ matrices, so if we deem the results not up to our desired accuracy, we can easily go to the second order approximation showed here, and only require calculation of one additional matrix $\mathcal{J}$.

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
