# Peer review of "NISQ Algorithm for Hamiltonian Simulation via Truncated Taylor Series"

_SciPost Physics, doi:SciPost Phys. 12, 122 (2022)_

## Round 1 · Referee Report · Anonymous (Referee 1) · 2021-6-24

Report

In this paper, the authors propose an algorithm for computing the time evolution of quantum many body systems. The method is 'in a way' assisted by a quantum device, since expectation values of K-body operators are measured in the beginning of the algorithm and subsequently used to govern the time evolution on a classical computer.

The idea is based on a variational approach. The quantum state at time $t$ is expanded in the basis of cumulative K-moment states, where the expansion coefficients act as variational parameters. This basis, even though not mentioned by the authors, is particularly well suited for describing states that are time evolved for short times $\Delta t$, which is immediately obvious from the Taylor-expansion of the time-evolution operator. However, for large times it is absolutely unclear if the state can be efficiently expressed in such a basis. In general, I would assume that the number of basis states required to approximate the state up to given error grows exponentially in time. This fundamental issue is neither discussed nor mentioned by the authors. The examples discussed show no convincing evidence that this algorithm is of any practical use. For this reason (and a couple of other reasons discussed below) I cannot support publication of this work. In the following I will describe my main criticisms in detail.

-One of the main motivations of the authors seems to be to get rid of the quantum-classical feedback loop of standard variational approaches for time evolution. However, in doing so it seems to me that an exponential overhead has been introduced. In my opinion, the question the authors have to answer is the following: How does the number of required basis states $|\chi_i\rangle$ grow as a function of time, and at which point does it reach the dimension of the Hilbert space?

I heavily criticize the way the cumulative $K-$moment states are introduced. The authors do not connect them to the Taylor expansion of equation 3. It should be pointed out explicitly that if the Taylor expansion is applied to a state $|\psi\rangle$, the result is exactly a linear combination of cumulative $K-$moment states. At this point it becomes pretty clear that after long times it would be very difficult to generate a sufficiently expressible ansatz. Furthermore, the notation in equation (5) is sloppy. It should be made clear that the indices $i_K,\cdots ,i_2, i_1$ all run from $1$ to $r$. It is not mentioned that the number of basis states in this set is exponential in $K$.

  • The authors test their approach on a couple of trivial examples: 2-qubit models that can be solved with pen and paper and a "classical" 4-spin example. Furthermore they investigate time evolution of an 8-spin Ising chain. According to Figure 3, for $K=1$ and $K=2$ the fidelity drops quickly to zero (as expected) while the $K=3$-moment expansion is able to capture the time evolution exactly. The Hamiltonian in equation (21) consists of 15 Pauli strings. If I count correctly, the number of basis states in the $K=3$ moment expansion is given by $15^3 + 15^2 + 15 + 1 = 3616$, which by far exceeds the Hilbert space dimension of $2^8 = 256$. So it seems to be no surprise at all that this is able to capture the dynamics. None of this is mentioned by the authors.

  • What is exactly the role of the initial state $|\psi\rangle$? The authors talk about an "efficiently preparable quantum state". What happens if this a product state in the computational basis? In this case, it seems to me, the matrices $\mathcal{E}$ and $\mathcal{D}$ can just be calculated with pen and paper. At this point, there is no need for a quantum computer/simulator. So apparently, the algorithm proposed by the authors is only meaningful if the initial state $|\psi\rangle$ is a nontrivial (possibly highly entangled) state. Do the authors have any specific application in mind? Again, none of this is discussed.

  • Is the algorithm as outlined on the right column of page 4 actually correct as its written down? In point 2, the authors say that $\mathcal{E}$ and $\mathcal{D}$ are measured for a fixed $K > 0$ and the job of the quantum computer is done. At stage 3 it is said, that if the fidelity is not satisfying, $K$ has to be increased. But if $K$ needs to be increased, $\mathcal{E}$ and $\mathcal{D}$ get larger and new measurements have to be taken. So the job of the quantum computer is not done, or do I misunderstand something?

In conclusion, I disagree with many of things put forward in this manuscript. The authors have to explain why this is an efficient algorithm and where a possible advantage from using a quantum-device comes in. In my opinion, the authors should reconsider their views on hybrid quantum-classical algorithms. The authors claim that a hybrid quantum-classical feedback loop is inefficient since the quantum computer has to wait for the output of the classical computer. In reality, it actually turns out that very often the opposite is the case. This is particularly true for AMO systems, where the repetition rate of the quantum machine is not particularly high but expectation values can be obtained with very high quality.

---

## Round 1 · Referee Report · Anonymous (Referee 2) · 2021-6-26

Report

In this paper the authors propose a hybrid quantum classical algorithm to calculate time evolution of many body quantum systems called truncated Taylor quantum simulation. They combine the idea of approximating the time evolution operator with a truncated Taylor series with systematically generating an ansatz state as a superposition of cumulative K-moments states, and considering the coefficients of the superposition as variational parameters to be classically optimized.

While the paper presents a novel approach to exploiting NISQ devices to study real-time dynamics of quantum systems, and the algorithm is well explained, there are very important analysis and discussions that I found missing, and therefore I have the following comments for the authors to address before giving my final recommendation.

1) There is no discussion as to why a superposition of cumulative k_moments states, although they are generated based on the Hamiltonian, is a suitable ansatz for the time-evolved state. In particular, I consider necessary to clarify bounds for the size of the set CS_K, and if there is any estimate for how high K needs to be for the ansatz to be expressive enough. This is also related to the claim that this algorithm is “exceptionally faster than the feedback loop based NISQ algorithm for simulating quantum dynamics”, which I do not find sufficiently justified.

2) In the thirds step of the description of the algorithm Δt is chosen based on knowledge of the eigenvalues of H, however diagonalizing a many-body Hamiltonian is a challenging task in its own right, and this knowledge should not be assumed.

3) A proposal to include higher orders in the power expansion of U(Δt) should be accompanied with a warning that more quantum resources are needed, if not a full discussion about the scaling of such an increase. The inability to simulate long timescales is mentioned in the text as one of the drawbacks of currently available algorithms, how does the proposed algorithm compare?

4) Although error mitigating techniques are mentioned for the data obtained through IBM, no error bars are shown in any of the plots, is it because they are too small to be shown? If so, this should be explicitly mentioned.

5) It is shown in figures 2 and 3 that a choice of K = 3 saturates the fidelity even for very long times for the respective models. This is a striking feature that is not discussed in detail. Should one expect such a saturation to happen for more general models? Under which conditions? Or is it a consequence of the simplicity of the considered models?

6) The data in Table 1 is very confusing. According to the notation in equation (5), for K = 1, S_K has r elements (as many as Pauli strings in H). Are the values of K wrongly labelled, or is equation (5) wrong? Further, how can increasing K from 3 to 4 for the 4 quit case increase the number of basis states only by one?

Other minor comments follow.

  • The indices in equation (5) seem to be incorrectly formatted.
  • Reference [3] has the name of the collaboration incorrectly formatted.
  • Typo in equation (21), the summation goes over 9 qubits, instead of 8.
  • Typo in the summation indices of equation(22).
  • Typo in Appendix A, end of first paragraph: “We” is repeated.

---

## Round 2 · Referee Report · Anonymous (Referee 2) · 2021-11-9

Report

The authors have addressed most of my comments and have improved on the previous version of the manuscript by fixing the bigger drawbacks. However, I think there is still room for better clarity and completeness.

1) The response to point 2 in my previous report is not reflected in any changes in the manuscript. 2) The authors have not replied to point 5 in my previous report. 3) A similar comment to the previous point applies to the results in fig 5. K = 3 performs remarkably well, this is a feature that should be discussed. 4) The clarity of fig. 5 could be improved by using different markers for different lines. 5) Some formal inaccuracies that I pointed out in the previous version still persist, please fix them. 6) Typo in the generalised eigenvalue equation. 7) A comment is misplaced above eq. (23).

---

## Round 2 · Referee Report · Anonymous (Referee 1) · 2021-11-25

Report

Overall, the authors took great efforts to answer my questions and to address my comments. My main criticism was related to the dimension of the variational space (i.e. the number of cumulative $k$-moment states) required to faithfully describe the time-evolved state $|\psi(t)\rangle$. To some extent, I do agree with the authors that this is related to the fundamental question of expressibility of variational quantum circuits. I would not go so far and say that this is in general a completely open question for all variational quantum algorithms. Most likely it is true when it comes to time evolution of many body systems. On the other hand, if one (for instance) uses a variational quantum eigensolver to prepare ground states of local gapped Hamiltonians, we know that these states should fulfil the area laws of entanglement and are thus described by finite bond-dimension tensor networks. According to that, we know that such states are described by a number of parameters that only scale polynomially in the system size, in which case a "relatively" short depth quantum circuits should provide sufficient expressibility. Time evolution as discussed in this manuscript is of course a different story. Nevertheless, I would add the statement that this fundamental problem of expressibility does particularly emerge in variational algorithms for time evolution.

I have the feeling that my question of how quickly the required size of the $k$-moment state-set grows as a function of time has not really been answered. The authors say that in the worst case, the parameter $K$ is equal to the rank of the Hamiltonian, but that means that in this worst case the method is impractical. I do not understand the statement of the authors: "... Thus, the number of states that we require in our Ansatz scales linearly with the rank of the Hamiltonian. We believe that this scales favourably compared to other NISQ algorithms such as VQS and VFF." How can this clearly exponential scaling be favourable compared to something else?

I still have the feeling that in the cases where the numerical results match the exact time evolution (for example Figure 3 a)), the number of basis states matches (or exceeds) the Hilbert space dimension. The authors pointed out that in this case there are 137 states in the set while the \textit{full} Hilbert space dimension is $2^8=256$. But of course the Ising model studied here exhibits certain symmetries, like reflection around the center or a global $Z_2$-symmetry which is perhaps (?) satisfied by the ansatz. These symmetries might easily reduce the dimension by a factor of 2.

The authors draw several connections to Krylov time evolution. In these algorithms the task is to apply $e^{-i \Delta t H}$ at each time step to the current state. To this end one constructs the Krylov subspace at every time step, based on powers of $H$ applied to the state. The number of Krylov vectors is related to the size of the time step one is able to perform. It seems to me that the algorithm proposed by the authors performs a single Krylov-timestep from the initial state. Thus, the maximum time that can be reached might be very limited.

To summarize, I am still concerned that the authors are overselling their approach to a certain extent. In my opinion, it is very important to point out clearly the capabilities of proposed quantum algorithms. In particular to elaborate on perspectives what these algorithms are able to achieve what classical computers cannot. For me, in the present case the answer is: With this algorithm one can time evolve a highly entangled state (a state that cannot be stored efficiently on a classical computer) for a short amount of time. The quantum device has no role here if the initial state is a product state. In this case, everything becomes classical. This point should be communicated very clearly.

Requested changes

I would ask the authors to point out clearly the framework in which their algorithm has meaningful applications. In my opinion this is the following: If a quantum device prepares a highly entangled state, i.e. a state that is difficult to store classically, this algorithm can be used to evolve such a state for a short period of time.

Alternatively, one could provide a detailed analysis on how many basis states are required as a function of time in order to approximate the state to a given fidelity. At the moment the authors say that in the worst case the number of basis states matches the rank of the Hamiltonian. But at this point, the algorithm is impractical.

---

## Round 2 · List of Changes

Changes are in blue in the PDF file submitted

---

## Round 3 · Referee Report · Anonymous · 2022-2-15

Report

I read the revised manuscript and in particular the changes made by the authors. My reamining questions and comments have been adressed. In particular the paragraph on page 7 adresses now limitations in terms of reachable time scales, which was one of my main concerns.

Also other questions like the connection of the proposed method to Krylov time evolution have been clarified.

I thus have no further objections to publication of this manuscript.

---

## Round 3 · Author Response

We would like to thank you again for considering our submission titled “NISQ Algorithm for Hamiltonian
Simulation via Truncated Taylor Series” for publication in Scipost. We would also like to extend our gratitude
to the referees for their second round of valuable feedback and reports. We have addressed their comments, point by point, in
the manuscript submitted to scipost below, and edited the manuscript accordingly. We hope that the improved manuscript is fit for publication in Scipost.

---

## Round 3 · List of Changes

We have given a point-by-point reply to the questions in the manuscript submitted to scipost.

---

## Editorial Decision

published